# Knowledge, Attitudes, and Subjective Norms Associated with COVID-19 Vaccination among Pregnant Women in Kenya: An Online Cross-Sectional Pilot Study Using WhatsApp

**DOI:** 10.3390/ijerph21010098

**Published:** 2024-01-16

**Authors:** Sylvia Ayieko, Walter Jaoko, Rose Okoyo Opiyo, Elkanah Omenge Orang’o, Sarah E. Messiah, Kimberly Baker, Christine Markham

**Affiliations:** 1Department of Health Promotion and Behavioral Sciences, The University of Texas Health Science Center at Houston School of Public Health, Houston, TX 77030, USA; kimberly.baker@uth.tmc.edu (K.B.); christine.markham@uth.tmc.edu (C.M.); 2Department of Medical Microbiology, University of Nairobi, P.O Box 19676, Nairobi 00202, Kenya; wjaoko@kaviuon.org; 3KAVI-Institute of Clinical Research, University of Nairobi, P.O. Box 19676, Nairobi 00202, Kenya; 4Department of Public and Global Health, University of Nairobi, P.O. Box 19676, Nairobi 00202, Kenya; roseopiyo@uonbi.ac.ke; 5Department of Reproductive Health, Moi University, P.O. Box 3900, Eldoret 30100, Kenya; bworango@mu.ac.ke; 6Department of Epidemiology, Human Genetics and Environmental Sciences, The University of Texas Health Science Center at Houston School of Public Health, Dallas, TX 75207, USA; sarah.e.messiah@uth.tmc.edu; 7Center for Pediatric Population Health, University of Texas Health Science Center at Houston School of Public Health, Dallas, TX 75207, USA; 8Department of Pediatrics, McGovern Medical School, Houston, TX 77030, USA

**Keywords:** COVID-19 vaccination coverage, pregnancy, maternal health, vaccine hesitancy, WhatsApp, Kenya

## Abstract

COVID-19 vaccination during pregnancy has been recommended, but the perceptions related to uptake remain unexplored. This pilot study aimed to explore how perceptions influence COVID-19 vaccine uptake among a sample of 115 pregnant women in Kenya, recruited via WhatsApp. Data were collected using an adapted online questionnaire between May and October 2022. Logistic analyses assessed the relationship between COVID-19 vaccination uptake and the Theory of Reasoned Action (TRA) constructs: attitudes and subjective norms. COVID-19 vaccination coverage was 73%, with vaccine hesitancy estimated at 41.4% among the unvaccinated group. Most participants had completed college education and had good knowledge of COVID-19 vaccines. There was no significant effect of enrollment in WhatsApp pregnancy groups on attitudes toward COVID-19 vaccination. Pregnant women were concerned about vaccine effectiveness (31.1%), and almost one-half (47.3%) were discouraged from receiving COVID-19 vaccines. Positive attitudes towards vaccination were associated with COVID-19 vaccination (aOR 2.81; 95% CI 1.12–7.04; *p* = 0.027), but no significant relationship was found between COVID-19 vaccination and strong subjective norms (influences to get COVID-19 vaccines). Our findings suggest that strategies to improve vaccination should consider targeting attitudes and proximal social networks (friends/family) to facilitate vaccination decision-making. WhatsApp can be used for research distribution and enhance the dissemination of accurate information.

## 1. Introduction

Vaccination is an established public health tool for preventing many infectious diseases and associated mortalities [1,2]. This is because inoculation processes are relatively safe, do not require frequent user intervention, and can ensure that large populations are protected from diseases [2]. Vaccination against the coronavirus disease 2019 (COVID-19) has prevented thousands of deaths and minimized hospitalizations due to severe COVID-19 infections [3,4].

The literature suggests that pregnant women were more susceptible to severe morbidities and mortality from COVID-19 compared with non-pregnant women of reproductive age [5,6]. A longitudinal study conducted in Western Kenya in 2020 revealed that 6% of pregnant women were diagnosed with COVID-19, compared to 4% of postpartum women [7]. Despite the risk, there are limited data on COVID-19 vaccination coverage and uptake during pregnancy in lower-middle-income countries [1], with vaccination rates ranging from 14.4% in Ethiopia [8] to 37.7% in Turkey [9]. Some possible explanations related to the lower rates of COVID-19 vaccination uptake include the lack of information on the safety of COVID-19 vaccines during pregnancy [10], vaccine hesitancy among healthcare providers [11], low susceptibility to COVID-19 [12], and concerns about rapid vaccine development [13].

GeoPoll’s study found that about 61% of Kenyans believed the COVID-19 vaccine was safe and effective [14]. Kenya participated in the COVID-19 Vaccine Global Access (COVAX) partnership program [15] with the goal of vaccinating the entire adult Kenyan population by June 2022 [16]. By May 2022, about 30% of the entire population in Kenya had received full vaccination [16]. Empirical evidence suggests minimal COVID-19 uptake can be attributed to environmental influences such as the unavailability of vaccines [10], weak health systems [17], and lack of trained personnel [1]. Societal influences such as misinformation and reduced community engagement have also been reported as reasons behind the low rollout of COVID-19 vaccination [6,13]. Policy changes and recommendations for COVID-19 vaccination during pregnancy [18,19,20] did not translate to higher uptake among pregnant women due to conflicting messages and limited information [21].

While a few studies have investigated attitudes of acceptance and hesitancy around COVID-19 vaccination among pregnant women [22,23], research suggests that comparatively few studies have assessed the role of psychosocial and behavioral determinants on COVID-19 vaccination in Sub-Saharan Africa [24,25]. To develop more effective intervention protocols, an in-depth understanding of the beliefs that guide vaccination is necessary to optimize public health efforts. This study used the Theory of Reasoned Action (TRA) as the theoretical framework [26]. The TRA’s constructs of attitudes and subjective norms (exposures) were examined to understand how they are associated with COVID-19 vaccination uptake (outcome variable). Attitudes are identified as the best predictors of intentions to perform a behavior and are significant in designing TRA-based interventions [27]. Subjective norms are operationalized as the vaccination beliefs of friends, family, or health providers influencing COVID-19 vaccination behavior.

Understanding the psychosocial influences of COVID-19 uptake in Kenya offers potential insight into targeted interventions to reduce vaccine hesitancy among women in Kenya. This pilot online cross-sectional study aimed to explore how knowledge and perceptions (attitudes and subjective norms) influenced COVID-19 vaccine uptake among pregnant women in Kenya using WhatsApp mobile app technology [28] as a recruitment tool.

## 2. Materials and Methods

### 2.1. Study Design, Setting, and Participants

This cross-sectional pilot study was conducted among pregnant women in Kenya. Convenience sampling was used to recruit participants from antenatal clinics at the Moi Teaching and Referral Hospital (MTRH) in Uasin Gishu and Kenyatta National Hospital (KNH) in Nairobi, which are the leading national referral hospitals in Kenya. Study participants were also recruited within WhatsApp groups that are created specifically for pregnant women. This is because the potential participants were already connected with the antenatal clinics as patients seeking care and were also familiar with using digital platforms, such as WhatsApp groups, for communication and interaction. With recent data indicating that WhatsApp can be utilized as an effective tool in social networking and data collection [29], digital posters about the survey were circulated via WhatsApp groups specifically created for pregnant women in Kenya.

The researchers for this study limited patient interactions, due to the COVID-19 pandemic, by recruiting pregnant women using WhatsApp and administering an online survey. Only participants who had access to WhatsApp were recruited. The estimated sample size of 100 (50 from each hospital) for the pilot study was based on the number of pregnant women expected to attend prenatal care during the study period. Given the study would not be representative of the entire population and the lack of published studies on COVID-19 vaccination uptake among pregnant women in Kenya, the sample size was estimated based a 10% margin of error using the following formula:*n* = *z*^2^ × *p* × (1 − *p*)/*e*^2^
where *z* = 1.96 for a confidence level (α) of 95%, *p* = proportion (expressed as a decimal), and *e* = margin of error. *z* = 1.96, *p* = 0.5, and *e* = 0.1 [30].

The eligibility criteria included pregnant women in Kenya, aged between 18 years and 49 years, and with access to WhatsApp and the internet to complete the online questionnaire. Participants were required to complete an eligibility screener by clicking on the Bitly link or the QR code on the digital flyer sent via WhatsApp to participate in the study. Once screened as eligible, participants were directed to the information and consent page, and only those who provided informed consent were automatically routed to the online survey. The electronic questionnaire was delivered, and data were stored within Qualtrics^®^ (Seattle, DC, USA) [31]—a secured web-based application. We ensured participant confidentiality and privacy by conducting the study online. Participants provided contact information only if they wanted to receive remuneration for completing the survey (300 Kenya Shillings). Participants had the option to terminate the survey at any time to minimize bias. Data were securely stored and transferred to an Excel spreadsheet stored on a password-protected computer, with only research team members having access to this data file.

### 2.2. Ethical Approval

The study protocol, recruitment material, data collection instruments, and informed consent information were approved by the university-affiliated institutional review boards in the United States (HSC-SPH-21-0997) and Kenya (MTRH/MU IREC/092/2022 & KNH-UoN P98/02/2022). Approval to conduct research was also granted by the National Commission for Science, Technology, and Innovation (NACOSTI) in Kenya. Researchers followed research procedures according to the institutional guidelines. Survey data were collected between May 2022 and October 2022.

### 2.3. Measures

Participants completed an online adapted questionnaire that included questions from the National Immunization Survey-Adult COVID Module (NIS-ACM) and the Omnibus survey [32]. The NIS-ACM survey, developed by the Centers for Disease Control and Prevention (CDC), has been used in numerous studies in the United States that have been published. The adapted questionnaire was validated by public health experts in Kenya and pre-tested by six women of reproductive age: two recent African immigrants to the United States and four women in Kenya. The demographic data collected included age, educational level, employment, insurance status, maternal vaccine history, prior COVID-19 infection, and enrollment in pregnant women’s WhatsApp groups. The study also collected data on pregnant women’s knowledge, attitudes, and subjective norms about COVID-19 vaccination. The outcome variable was COVID-19 vaccination status in response to the question, “Have you received at least one dose of a COVID-19 vaccine?” The independent variables were knowledge, attitudes towards vaccination (safety, importance of the vaccine), subjective norms (perceived number of vaccinated friends/family; influences to receive COVID-19 vaccines) and intentions. For knowledge, participants responded to the statement, “COVID-19 vaccination can help control the spread of COVID-19 disease” (True or False). An example of an attitude question was “How important do you think the COVID-19 vaccine protects you against COVID-19?” while a question related to subjective norms was “Which of the following tried to influence you to get a COVID-19 vaccine?” Subjective influences were categorized as proximal (friends/family/healthcare providers), and distal (religious leaders, government, celebrities). The survey items were assessed using four-point Likert scales ranging from Very Strongly Agree—4 to Do not Agree—1 or Very Concerned—4 to Not Concerned at All—1. For the logistic analysis, independent variables were recoded as dichotomous variables; the demographic data were retained as categorical variables.

### 2.4. Statistical Analysis

Descriptive statistics were used to characterize distributions of demographic data and COVID-19 vaccination coverage. We also summarized the results from participants’ responses on COVID-19 vaccination knowledge, attitudes, and subjective norms. Bivariate and multivariable logistic regressions examined the relationship between COVID-19 vaccination uptake and participant perceptions of COVID-19 vaccination, reported as odds ratios with 95% confidence intervals (CIs). In the multivariable model, adjusted odd ratios (aORs) were calculated for COVID-19 vaccination uptake, with the TRA constructs as independent variables while controlling for age and level of education. Moderation analyses were conducted with enrollment in WhatsApp pregnancy groups as a moderator. The statistical software for data science (STATA v.18) [33] was used for the statistical analysis. In all analyses, *p* < 0.05 was considered statistically significant.

## 3. Results

A total of 174 participants attempted the survey. The final sample comprised *n* = 115 pregnant women who met the eligibility criteria and were willing to participate. Of the 115 responses, missing data were less than 3%; hence, imputations were unnecessary. See Figure 1.

### The Sociodemographic Characteristics

The sociodemographic characteristics of pregnant women are provided in Table 1. Almost all the participants accessed the survey using the WhatsApp link, with only two using a QR code from the recruitment flyer distributed. Almost three-fourths of the pregnant women (73.0%) had received at least one dose of COVID-19 vaccination by October 2021. Of those who had received COVID-19 vaccination, 86.9% reported receiving the full dose of COVID-19 vaccines (One dose of the Johnson and Johnson COVID-19 vaccine and two or more doses of the other COVID-19 vaccines). Most pregnant women were between 25 and 29 years (39.1%) and 30 and 39 years (39.1%). The majority of the study participants had a high school degree or higher, and 45.5% were employed in businesses/institutions that mandated COVID-19 vaccination. Less than one-third (28.7%) of the study participants reported being in WhatsApp groups for pregnant women. About three-quarters of the study sample reported having health insurance, 18.3% had a comorbid health condition, and less than half (41.7%) of the pregnant women had a previous COVID-19 infection.

Most study participants (97.4%) had accurate knowledge of COVID-19 vaccination (Table 2). The perceptions about the COVID-19 vaccination during pregnancy revealed that most (83.8%) of the pregnant women in Kenya were confident that the vaccine was safe; however, a third did not believe that COVID-19 vaccines could protect against COVID-19. In comparison, 80% felt responsible for getting vaccinated against COVID-19. When asked about people who influenced pregnant women to get COVID-19 vaccines, slightly more than half the participants (56.5%) reported that they were encouraged by friends or family, followed by doctors/healthcare providers and co-workers. About half of the study participants (47.3%) reported being discouraged or turned away from vaccination while pregnant. Among the unvaccinated group, 58.6% intended to get vaccinated in the future.

The statistical analysis showed that the adjusted odds of vaccination were three times higher among the older pregnant women (30 years or older), and the relationship was statistically significant [aOR-3.38; CI 1.19–9.60]. COVID-19 vaccination was significantly associated with positive attitudes, where pregnant women reporting higher attitude scores were three times more likely to be vaccinated [aOR-2.96; CI: 1.20–7.29] compared to those with negative attitudes. However, subjective norms, knowledge, or enrollment in WhatsApp pregnancy groups were not associated with COVID-19 uptake among pregnant women. See Table 3 for additional information.

A moderation test (WhatsApp pregnancy group enrollment as a moderator) with attitudes as an independent variable and COVID-19 vaccination as the outcome variable showed there was a significant main effect found between attitudes and COVID-19 vaccination, *b* = 0.98, CI [0.09, 1.88], *p =* 0.03, and nonsignificant main impact of WhatsApp pregnancy group enrollment on COVID-19 vaccination, b = 1.31, CI [−2.13, 4.74], *p* = 0.46. The interaction between attitudes, WhatsApp pregnancy group enrollment, and COVID-19 vaccination was nonsignificant. Table 4 shows additional results from the moderation analysis. 

## 4. Discussion

The present study assessed the knowledge, attitudes, and subjective norms of pregnant women in Kenya about COVID-19 vaccination during pregnancy. While pregnant women were considered at risk for severe COVID-19 infections, knowledge about COVID-19 vaccines and influence from significant people to get vaccinated did not necessarily translate to COVID-19 vaccination uptake. The study outcomes showed that positive attitudes towards COVID-19 vaccination had a substantial impact on vaccine uptake, with almost the entire sample (97.4%) having accurate knowledge of COVID-19 vaccines and majority of the study sample (73.0%) reporting COVID-19 vaccination uptake. Even among the unvaccinated, more than half (58.6%) intended to get vaccinated. There was higher confidence in vaccine safety (83.8%) compared to effectiveness (68.7%). While friends/family members and healthcare providers tried to influence pregnant women to get vaccinated against COVID-19, the role of these close social networks were not significant. Despite fewer pregnant women (28.7%) reporting being in WhatsApp groups specifically for pregnant women, the lack of enrollment did not influence COVID-19 vaccination uptake. The study findings demonstrate the importance of addressing perceptions of vaccination as a public health strategy, the implications for research in behavioral interventions, and the interplay between community engagement and patient–provider communication with pregnant women in Kenya.

As a pilot study, we aimed to investigate WhatsApp use in recruitment and research among pregnant women in Kenya. The study personnel reported data collection challenges, probably because not all pregnant women had smartphone access to WhatsApp. Approximately one-third of the 174 people who attempted the survey were ineligible, with about 21 (12%) eligible participants not consenting. An estimated 18 million people in Kenya use the internet, with 41% utilizing the WhatsApp platform as their favorite social media channel [28], suggesting that not all pregnant women felt comfortable using the messaging application. Other studies have reported negative psychological associations among WhatsApp users exposed to COVID-19-related information [34,35]. While some clinical staff encouraged pregnant women to participate in the study, using an online survey reduced any coercion likely to occur in public hospitals if women perceived that they could receive inferior services if they declined to participate. In the past, data in Kenya have been collected by researchers who engaged women in person and input the data directly on paper or iPads [36]. A study by Jacaranda Health also indicated the feasibility of using WhatsApp to communicate with pregnant women in Nairobi [37]. With the increasing use of WhatsApp messenger as an official channel of information [34] and other recent innovations, such as polls included in the application, health programs could utilize WhatsApp for advanced mechanisms for research and data collection in the future.

The second aim of our study was to determine COVID-19 vaccination coverage among pregnant women in Kenya. Our study indicated that over 73% of pregnant women had already received at least one dose of the COVID-19 vaccine, which was much higher than the 28% previously reported in Kenya [38] and even in the general Kenyan population [16]. The higher vaccination rates in this study sample could be due to shifts in policy recommendations for COVID-19 vaccination [18,19,20], more availability of coronavirus disease 2019 vaccines through the COVAX initiative [15,16], and the sample characteristics—higher education levels, pregnant women seeking prenatal care in health facilities, and the setting (participants from referral hospitals). In the United States, about 55.4% of pregnant persons had received the COVID-19 vaccine by July 2021, despite the availability of COVID-19 vaccination and recommendations from the Centers for Disease Control and Prevention (CDC) [22,39]. Besides vaccine access, political, historical, and cultural factors could influence vaccination among pregnant populations besides the availability of vaccines [40].

The opinion that COVID-19 vaccination is not safe during pregnancy was held by 16.2% of our study participants, although a higher proportion (68.7%) of the women believed that COVID-19 vaccination could protect them from COVID-19. Similar to a study in Ethiopia [8], our study showed that most pregnant women had positive attitudes towards COVID-19 vaccination, and attitudes were significantly associated with COVID-19 vaccination. The scarcity of data on the safety of the COVID-19 vaccine during pregnancy [5], especially earlier in the pandemic, may have contributed to higher vaccination hesitancy. Reviews on COVID-19 vaccination during pregnancy also reported attitudes as factors influencing vaccine acceptance [21,24].

To our knowledge, this is among the few studies that examined subjective norms as a possible factor related to COVID-19 vaccination in pregnancy [41]. Other studies in the general population that have reported an association between vaccination and subjective norms [42] suggested the role of family members’ support in immunization. While our study did not indicate the association between norms and COVID-19 vaccination, descriptive data showed that proximal networks (family members, friends, and healthcare providers) strongly encouraged pregnant women to get vaccinated against COVID-19, which is similar to prior research [43]. With pregnant women reporting greater proximal influences compared to distal social influences (government officials, religious leaders, or celebrities), interventions should equip families and healthcare providers with timely and accurate information to assist women in decision-making.

Consistent with our findings, other authors have reported good knowledge of COVID-19 vaccination among pregnant women [38]. Contrary to previous studies that showed a relationship between knowledge and COVID-19 vaccination acceptance [44], our study did not show a significant association between the two, as nearly all participants believed that COVID-19 vaccination could help control the spread of COVID-19 infections. Given that COVID-19 was an emerging disease, in 2020, scientists across the globe utilized the information available at the time to inform populations about the best ways to help control and prevent severe COVID-19 infections [11,45,46]. As more research is conducted on COVID-19 and related vaccinations, some public health prevention practices such as wearing face protection, hand washing, and vaccination may still be recommended as other treatment options become available. During the pandemic, public health agencies, governments, and international organizations disseminated substantial information about COVID-19 [11,45,46]. However, a plethora of misinformation circulating via social media impacted public health interventions [47]. Some scholars argue that the ‘balkanization’ of COVID-19 information during the pandemic and dis/misinformation likely contributed to vaccine hesitancy [40]. As such, there is a need for vaccination programs to shift from just targeting knowledge through awareness campaigns to deploying innovative approaches that influence other psychological processes among pregnant women.

Vaccine intention, sometimes defined as the willingness to get a COVID-19 vaccine, was 58.6% among pregnant women in this study who were not already vaccinated, which was higher than what has been previously reported in Kenya (49.2%) [48], Nigeria (8.4%) [49], and Ethiopia [50]. Since the unvaccinated group in our sample was small, we did not conduct any analysis with this sub-sample. A longitudinal study among pregnant and postpartum women in Kenya reported changes in vaccination willingness (38% to 71%, *p* < 0.001) within a 2-year timeframe [36], suggesting that various factors could influence vaccine hesitancy among populations. Despite data showing COVID-19 vaccine safety and efficacy during pregnancy [5], there have been only slight changes in vaccination intention rates [21], necessitating thorough and systematic investigative research on vaccine hesitancy.

In summary, our study demonstrated that while vaccine coverage in this sample was much higher than in the general population in Kenya, health programs should consider interventions that address attitudes around preventative maternal behaviors, including vaccinations. In terms of research, the authors acknowledge the need to include additional questions, such as the intentions for full vaccination dosage among partially vaccinated pregnant women, for a more comprehensive outlook on vaccination behaviors. Public health action plans should also expand health communication campaigns that equip healthcare providers and the community with timely and accurate information. The most recent data show that about 30% of the general population in Kenya have received complete COVID-19 vaccine doses [16,51]. As such, the government and the Ministry of Health in Kenya must implement strategies to protect pregnant women, their families, and the general population in future pandemics.

### Strengths and Limitations

Given the increasing rates of technology and internet use, this study demonstrated that leveraging available online platforms such as WhatsApp could be used for research and potentially to target interventions among pregnant women in Kenya. A theory-informed approach illustrated that attitudes are associated with COVID-vaccination uptake and confirmed some TRA assumptions. The model also highlighted the role of family, friends, and healthcare providers as critical influences for pregnant women and the need for more research on subjective norms. The information gathered in this study will also provide feedback to hospitals and healthcare providers. For example, although accurate knowledge of COVID-19 vaccination showed the effectiveness of awareness campaigns, vaccine hesitancy was still an issue in some groups despite higher education levels.

In addition, the use of the NIS-ACM questionnaire developed by the CDC, a tool that has been used widely in the US, guided this study. By utilizing similar questions but adapting them to the Kenyan context, this study highlighted the perceptions of COVID-19 vaccination among pregnant women in Kenya.

One of the pitfalls of this study is that we cannot effectively determine direct causal relationships between attitudes and COVID-19 vaccination, especially for pregnant women who have already received the COVID-19 vaccine. The inability of the study to establish temporal precedence [52] makes it problematic to understand if the attitudes of participants led to COVID-19 vaccinations or if the participants changed their attitudes after getting vaccinated. A prospective longitudinal study would be a more prudent approach, but such a process is expensive, time-consuming, prone to risk of loss of follow-up of subjects who may drop out of the study [53], and may not be feasible with late antenatal care initiation reported in some regions in Kenya [54,55]. Other limitations of this study include the small sample size and the possibility of confounder bias since the participants were not randomized [52]. The study sample was not representative, given the higher rates of participants with a college degree or higher, which likely created a bias; thus, the findings should be taken with caution. However, as a pilot study, we attempted to reduce variation from potential confounders by restricting the sample to only pregnant women. We stratified the sample by age groups, education level, and enrollment in WhatsApp pregnancy groups.

Despite these limitations, this pilot cross-sectional study provides a snapshot of the prevalence of COVID-19 vaccination among pregnant women seeking care at national hospitals in Kenya, and the data collected will be critical in future studies on maternal immunization. The study also confirmed findings from prior studies that WhatsApp is an effective tool for recruitment and research data collection [56]. Subsequent studies of pregnant women should also explore using more representative samples by collecting data from diverse health facilities and communities.

## 5. Conclusions

While vaccine coverage in this sample was much higher than in the general population in Kenya, health programs should consider interventions that address attitudes around preventative maternal behaviors, including vaccinations. Public health action plans should also expand health communication campaigns that equip healthcare providers and the community with timely and accurate information. Findings from the study suggest that research among pregnant women in lower-middle-income countries utilizing online platforms and mobile apps is feasible, but recruitment should also incorporate traditional engagement approaches with participants. Future studies should consider using qualitative study methods to explore further the issues that arise from conducting quantitative studies in this population.

## Figures and Tables

**Figure 1 ijerph-21-00098-f001:**
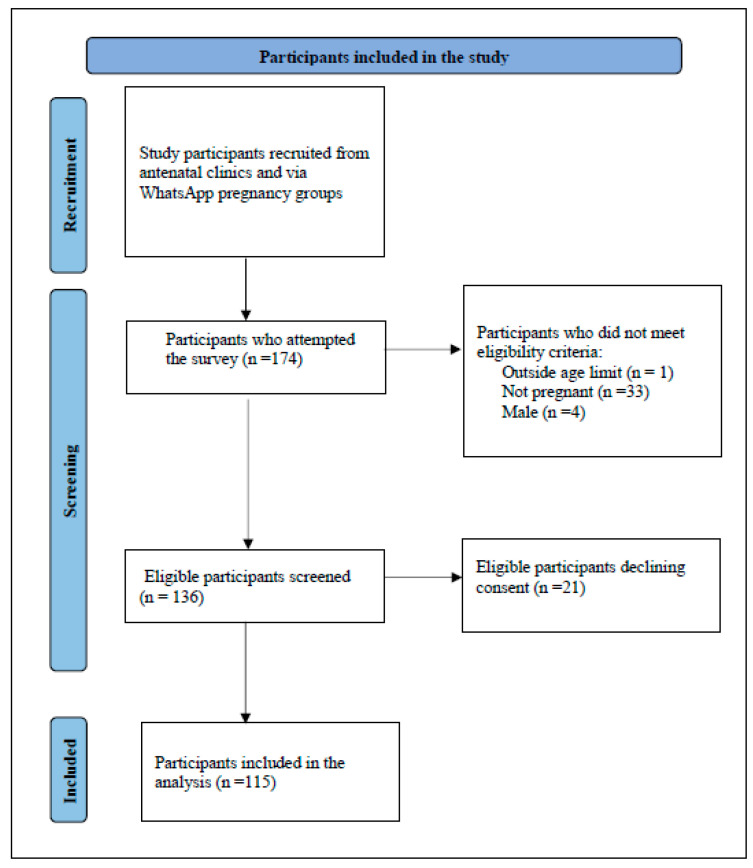
Flowchart of study participants enrolled in a pilot study on perceptions of COVID-19 vaccination among pregnant women.

**Table 1 ijerph-21-00098-t001:** Descriptive statistics of sociodemographic variables among a sample of pregnant women in Kenya (*n* = 115).

Variable	Overall *n* (%)
Pregnant women	115 (100)
Vaccination Status	
Vaccinated	84 (73.0)
Fully vaccinated	73(86.9)
Partially vaccinated	11(13.1)
Unvaccinated	31 (27.0)
Age	
18–24 years	19 (16.5)
25–29 years	45 (39.1)
30–39 years	45 (39.1)
40–49 years	6 (5.2)
Education Level	
Primary school	3 (2.6)
Secondary school	34 (29.6)
College/University	78 (67.8)
Workplace Vaccine Requirements	
No	32 (28.6)
Yes	51 (45.5)
Unemployed	29 (25.9)
Region	
Nairobi	59 (51.3)
Uasin Gishu	56 (48.7)
WhatsApp Pregnancy Group	
No	82 (71.3)
Yes	33 (28.7)
Insurance Status	
No	30 (26.1)
Yes	85 (73.9)
Comorbid Conditions	
No	94 (81.7)
Yes	21 (18.3)
Previous COVID-19 Infection	
No	67 (58.3)
Yes	48 (41.7)

**Table 2 ijerph-21-00098-t002:** Knowledge, attitudes, and subjective norms related to COVID-19 vaccination among a sample of pregnant women in Kenya.

Variables		*n* (%)
Knowledge	Knowledge of COVID-19 vaccination	
	Yes	112 (97.4)
	No	3(2.6)
Attitudes	Confidence in vaccine safety	
	Yes	93 (83.8)
	No	18 (16.2)
	Confidence in vaccine effectiveness	
	Yes	79 (68.7)
	No	36 (31.3)
	Responsibility for getting vaccinated	
	Agree	92 (80.0)
	Disagree	23 (20.0)
Subjective norms	Influences to get COVID-19 vaccination ^a^	
	Friends or family	63 (56.3)
	Doctors/Healthcare workers	22 (19.6)
	Co-workers	22 (19.6)
	Employers	13 (11.6)
	None	12(10.7)
	Number of vaccinated friends/family:	
	Almost all/Many	65 (58.0)
	Some/None	47 (42.0)
	Discouraged from vaccination	
	No	59 (52.7)
	Yes	53 (47.3)
Intention *	Intention to get vaccinated	
	Yes	17 (58.6)
	No	12 (41.4)

^a^ More than one response could be given to this question. * Only asked to participants who reported “NO” to already being vaccinated.

**Table 3 ijerph-21-00098-t003:** Association between COVID-19 vaccination, attitudes, and subjective norms among a sample of pregnant women in Kenya.

Age	OR (95% CI)	aOR (95% CI)
18–24 years	2.5 (0.71–8.76)	3.49 (0.86–14.10)
20–29 years	Ref	Ref
30–39 years	3.11 (1.22–7.92) *	3.38 (1.19–9.60) *
Education		
Primary	0.17 (0.38–2.39)	0.26 (0.02–341)
Secondary	0.96 (0.01–2.01)	1.04 (0.39–2.82)
College/University	Ref	Ref
WhatsApp Pregnancy Group	1.54 (0.59–4.02)	1.80 (0.62–5.17)
Attitudes	2.42 (1.11–5.28)	2.81 (1.12–7.04) *
Subjective Norms	0.65 (0.35–1.22)	0.68 (0.34–1.36)
Knowledge	0.73 (0.06–8.37)	1.68 (0.08–36.30)

* Statistically significant findings *p* < 0.05. R = Odds ratio; aOR = Adjusted odds ratio.

**Table 4 ijerph-21-00098-t004:** Moderation analysis: attitudes, COVID-19 vaccination, and WhatsApp pregnancy group enrollment among a sample of pregnant women in Kenya.

			Confidence Interval		
Variables	Coefficient	Std Error	Lower Limit	Upper limit	*p*-Value
Attitudes	0.98	0.46	0.09	1.88	0.03 *
Pregnancy WhatsApp group enrollment	1.31	1.75	−2.13	4.74	0.46
Attitudes × WhatsApp pregnancy group	−0.58	0.99	−2.52	1.35	0.56

* Statistically significant finding *p* < 0.05.

## Data Availability

Raw data used in this study, including de-identified participant data and survey results, are available upon request.

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
