# Peer review of "Knowledge, Attitudes, and Subjective Norms Associated with COVID-19 Vaccination among Pregnant Women in Kenya: An Online Cross-Sectional Pilot Study Using WhatsApp"

_ijerph, 2024, doi:10.3390/ijerph21010098_

Round 1
Reviewer 1 Report
Comments and Suggestions for Authors
Thanks for incorporation with my suggestions, goodluck
Author Response
Comment: Thanks for incorporation with my suggestions, goodluck
Response: We appreciate your suggestions that have strengthened the quality of the paper.
Reviewer 2 Report
Comments and Suggestions for Authors
The pregnant women in the sample are very highly educated. From a descriptive standpoint, this findings of the study provide limited generalizeability.
I have concerns about the measurement of both the independent and dependent variables that can't be fixed without recollecting data. The "knowledge" variable is not described, but they record that 97% were "knowledgeable" about COVID-19 vaccines. First, I don't know if this is a subjective (i.e. felt knowledgeability) or something else. As there is a virtual unanmity among study participants on this variable, it will be unrelated to the dependent variable by default. The dependent variable is simply whether or not the woman has been vaccinated at least one time. It is unclear in the document whether 1 dose was the recommended dose such as the J and J vaccine or whether two or more shots were requried for full vaccine benefit. Additional questions on whether or not they intended to get future doses if recommended should have been included here. Then the researchers could assess the relationship between present attitudes and intentions for future behavior with regard to vaccines. As it is, we have a measure of past behavior and are relating it to present attitudes and subjective norms. It is true that the researchers collect vaccine intentions among the currently unvaccinated women, but as this group has a sample size of 29, one can't do meaningful statistical analysis on such a small group.
The researchers note that they can only identify correlations between current attitudes/subjective norms and the dependent variable, they can't say much about causation.
In total, the study doesn't give us much new information on either the descriptive level or in terms of variable relationships.
Author Response
Thank you for your time and valuable feedback. We have incorporated your suggestions in the manuscript. Below are responses to the comments. We have used track changes for revisions and have highlighted the corresponding changes in the revised manuscript in yellow.
Comment: The pregnant women in the sample are very highly educated. From a descriptive standpoint, this findings of the study provide limited generalizeability.
Response: We agree that a large percentage of pregnant women in this sample were highly educated, which is not representative of the general pregnant population in Kenya. This is highlighted as a limitation in lines 456-458 (highlighted)
Comment: I have concerns about the measurement of both the independent and dependent variables that can't be fixed without recollecting data. The "knowledge" variable is not described, but they record that 97% were "knowledgeable" about COVID-19 vaccines. First, I don't know if this is a subjective (i.e. felt knowledgeability) or something else. As there is a virtual unanmity among study participants on this variable, it will be unrelated to the dependent variable by default.
Response: We have added more details about how knowledge was measured.
- Knowledge was assessed by the following statement. See lines 162-163
”COVID-19 vaccination can help control the spread of COVID-19 disease (True or False).”
- We assessed knowledge on COVID-19 vaccination based on the publicly available information from the Ministry of Health in Kenya. Given the dynamic nature of COVID-19 and other prevention strategies, we used what was considered “objective” at the time as knowledge. We appreciate the continual research on COVID-19 and COVID-19 vaccinations, realizing that changes in vaccine efficacy can change what was initially considered primary prevention.
- We have included additional information about knowledge in the discussion section. See lines 391-397
“Given that COVID-19 was an emerging disease, in 2020, scientists across the globe utilized the information available at the time to inform populations about the best ways to help control and prevent severe COVID-19 infections. As more research is conducted on COVID-19 and related vaccinations, some public health prevention practices such as wearing face protection, hand washing, and vaccination may still be recommended as other treatment options become available.”
Comment: The dependent variable is simply whether or not the woman has been vaccinated at least one time. It is unclear in the document whether 1 dose was the recommended dose such as the J and J vaccine or whether two or more shots were requried for full vaccine benefit.
Response: We have included additional information about the vaccine dosage in lines 215-217 and also provided that information in the table.
“Of those who had received COVID-19 vaccination, 86.9% reported receiving the full dose of COVID-19 vaccines (One dose of the Johnson and Johnson COVID-19 vaccine and two or more doses of the other COVID-19 vaccines).”
Comment: Additional questions on whether or not they intended to get future doses if recommended should have been included here. Then the researchers could assess the relationship between present attitudes and intentions for future behavior with regard to vaccines. As it is, we have a measure of past behavior and are relating it to present attitudes and subjective norms.
Response: Thank you for pointing this out. Unfortunately, we did not include this question in the survey but have included it in the discussion section as a consideration for future research. Lines 421-423
“In terms of research, the authors acknowledge the need to include additional questions such as the intentions for full vaccination dosage among partially vaccinated pregnant women for a more comprehensive outlook on vaccination behaviors”.
Comment: It is true that the researchers collect vaccine intentions among the currently unvaccinated women, but as this group has a sample size of 29, one can't do meaningful statistical analysis on such a small group.
Response: We certainly agree that the unvaccinated group in this sample was small. As a pilot study, we did not anticipate such a high prevalence of vaccinated pregnant women in our sample. We would have analyzed vaccine intentions if we had a larger unvaccinated group.
Comment: The researchers note that they can only identify correlations between current attitudes/subjective norms and the dependent variable, they can't say much about causation.
Response: As a cross-sectional study, we cannot establish causality and have stated this as a study limitation. See lines 446-448.
Comment: In total, the study doesn't give us much new information on either the descriptive level or in terms of variable relationships.
Response: We respect your opinion about the study. However, we believe this study sheds light on COVID-19 vaccination among pregnant women in Kenya and also established that WhatsApp can be valuable in recruiting pregnant women in Kenya. The paper addressed subjective norms that have not been fully explored in Kenya.
As a pilot study, we have presented our findings and lessons learned that can be applied in future studies. Given the changes in technology and data collection, our research underscored the role of WhatsApp as a recruitment tool. We plan to share findings with the two referral hospitals and the Ministry of Health in Kenya to supplement current healthcare practices.
Round 2
Reviewer 2 Report
Comments and Suggestions for Authors
Nothing new to add